# RetFormer: Enhancing Multimodal Retrieval for Image Recognition

## Abstract

The expansion of Transformers and the collection of high-quality multimodal datasets have propelled deep neural networks to achieve unprecedented performance in vision and language tasks. However, applying these advances is non-trivial in real-world applications. The extensive number of parameters complicates model updates, and real-world data often features a long-tailed distribution along with noisy labels. To address the above issues, we propose to explore the internal structure of the neural network for learning with sample relationships, rather than just increasing the number of model parameters. Specifically, we introduce RetFormer, a model enhanced with a multimodal knowledge base for storing world knowledge, and a retrieval cross-fusion module designed to establish robust multimodal sample relationships by leveraging content from the knowledge base. RetFormer establishes a robust relationship between image and text modalities by integrating information from external knowledge bases into the model's decision-making process, thus overcoming the limitations of traditional approaches on model size and datasets. Our experiments demonstrate the benefits of integrating large-scale image-text datasets into vision tasks and exemplify the importance of modeling the relationship between image and text modalities. We have evaluated our approach on the task of long-tailed recognition and learning with noisy labels and have shown that it achieves state-of-the-art accuracies.

## 1 Introduction

The conjunction of large-scale Transformers with extensive pre-trained datasets has met with significant success in both vision and NLP domains. Large language models such as the PaLM series(Chowdhery et al., 2023; Anil et al., 2023), GPT series(Brown et al., 2020; Ouyang et al., 2022), and LLaMA series(Touvron et al., 2023a;b), along with multimodal large language models like Gemini(Team et al., 2023), GPT-4(Achiam et al., 2023), and Claude 3(Anthropic, 2024), have demonstrated state-of-the-art performance across various downstream tasks, owing to their robust capabilities in understanding, reasoning, and generation. According to the scaling law(Kaplan et al., 2020), extensive multimodal and multilingual datasets such as WIT(Srinivasan et al., 2021), LAION(Schuhmann et al., 2022), and DataComp(Gadre et al., 2024) have the same importance as large Transformers. Nevertheless, this approach is critically dependent on the collection of large-scale training data samples and the increase of model parameters, which turns out to be non-trivial in real-world applications.

In the training of large-scale Transformers, world knowledge is implicitly encoded within the vast number of model parameters, which can exacerbate certain challenges inherent in the current machine-learning paradigm. These challenges include difficulties in model updating, limited interpretability, and scalability issues. Furthermore, real-world datasets invariably encounter two principal problems: (a) noisy labels resulting from the ambiguity of the data itself and annotator errors, (b) class imbalance arising from natural phenomena. These issues often occur simultaneously, complicating the estimation of the true distribution of the dataset. The quest for robust deep representation learning through the exploration of sample relationships in scenarios of data scarcity or interference from noisy labels has garnered significant interest from the research community, particularly for tasks where there is a lack of high-quality training data to ensure generalization, such as learning with noisy labels and long-tailed recognition. While previous research(Zhang et al., 2023; Wei et al., 2021) efforts aimed at addressing these challenges have made significant contributions,

Figure 1: **Compare previous methods with ours and the knowledge contained in the different retrieved content.** (a) Previous methods focus only on image modalities and have a large number of model parameters, while (b) our method retrieves multimodal information from knowledge base. (c) and (d) give intuitive explanations for the correlations and differences between the image and text modalities. (e) describes useful knowledge that can be retrieved from the knowledge base.

most of them have been limited to image-centered solutions(Chawla et al., 2002; Khan et al., 2017; Wang et al., 2017), and the integration of retrieval-augmented into the field of vision recognition also remains an underexplored area. It remains a formidable challenge to devise a unified, flexible, and potent approach that can investigate the relationships among samples for robust representation learning without resorting to the simplistic strategy of merely increasing the number of model parameters.

As illustrated in Figure 1 (c)(d)(e), we search for images and the corresponding description text related to the query image. When we search for samples of the same class, the image modality presents concrete low-level invariant features (e.g., shape, color, texture), and the textual modality usually contains much high-level and abstractly relevant information. When we search for different classes of samples, there may be shared knowledge transfer in the image and text modalities with the query image. There may also be optional semantic information in the textual modality when we search for noise labels. In addition, textual descriptions are a priori knowledge that can be summarized by experts, which may be useful when there are not enough images to learn a general class representation for recognition.

To address the aforementioned issues, we propose an alternative perspective based on the above insight. As shown in Figure 1 (b), instead of statically compiling world knowledge into model weights, we construct an external image-text pair knowledge base for storing world knowledge and then utilize a retrieval module to identify and obtain relevant knowledge from a predefined knowledge base, modeling the effective relationships between image and text modalities, which is then used to enhance the model's predictive power. We overcome the limitations of traditional methods on model size and training datasets by seamlessly integrating information from external databases into the model's decision-making process. This semi-parametric approach enables the model to incorporate external knowledge to improve its understanding and predictive capabilities.

To evaluate our approach, we focus on long-tailed recognition and learning with noisy labels, which are challenging and meaningful tasks. We conduct extensive experiments on three datasets: CIFAR-100-LT(Cao et al., 2019b), ImageNet-LT(Liu et al., 2019b), and a real-world noisy dataset WebVision(Li et al., 2017).

In conclusion, our primary contributions are fourfold:

(1) We analyze and highlight the interplay and distinctions between image and text modalities within the context of retrieval augmentation in vision, noting that the text modality can complement retrieval content, which aids in long-tailed recognition and learning with noisy labels.

(2) We introduce a new image-text retrieval-augmented framework called RetFormer, which utilizes a small external knowledge base to model the relationship between text and image knowledge, thereby enhancing the performance of the model without incurring significant computational overhead.

(3) We validate RetFormer through extensive experimentation on CIFAR-100-LT, ImageNet-LT, and WebVision, demonstrating its superior performance over existing state-of-the-art methods.

(4) We provide an intuitive explanation for the effectiveness of this approach from a theoretical perspective based on gradient propagation.

## 2 RELATED WORK

**Data Scarcity Learning.** Learning with imperfect training data has proven to be a very challenging task and has been explored in a variety of data-scarce tasks. Long-tailed recognition is one of the key problems. Most of the solutions are variants of the core idea of "adjustment" (e.g., re-sampling(Chawla et al., 2002; Han et al., 2005; He & Garcia, 2009), re-weighting(Cui et al., 2019; Zhong et al., 2021)). Learning with noisy labels is another important task. The main methods revolve around filtering noisy labels, including but not limited to correcting wrong labels(Liu et al., 2022), reweighting examples(Ren et al., 2018), and selecting confident examples(Patel & Sastry, 2023; Wang et al., 2022). Some methods combine multiple techniques, e.g., DivideMix(Li et al., 2020) and Sel-CL+(Li et al., 2022).

**Sample Relationship.** There are diverse and tight relationships between different samples, and these relationships are widely used through various types of strategies.Mixup(Zhang et al., 2017) introduces prior knowledge to the model by performing simple linear mixing between training samples. Some approaches via investigating sample/class relationships to conduct transductive inference, e.g., transductive few-shot classification(Liu et al., 2018), and meta embedding(Liu et al., 2019a; Zhu & Yang, 2020). BatchFormer(Hou et al., 2022) applied a vanilla Transformer encoder into the batch dimension of each mini-batch to implicitly explore sample relationships during training.

**Retrieval augmented in computer vision** Recent approaches in computer vision perform various tasks by retrieving from external memory. Nakata et al. (2022) store feature maps from the training set in the memory, and perform k-NN for classification. RDM(Blattmann et al., 2022) retrieves nearest neighbors from a memory for generative vision models. RAC(Long et al., 2022) uses only the training dataset itself as an external source of information and extracts embeddings of relevant text segments with the pre-trained CLIP model. MAM(Iscen et al., 2023) considers texts to be of different importance from each other and introduces an additional dataset.

Our work is different in that we consider that both image and text modalities possess exploitable knowledge. Therefore, we propose to make the neural network itself capable of learning multimodal sample relationships based on the maximum inner product search algorithm. Experiments show better robustness of our novel framework across multiple tasks.

## 3 METHOD

### 3.1 PRELIMINARIES

We consider a supervised classification problem. The model has access to an imbalanced set of $N$ training samples $S = (x_i, y_i)_{i=1}^{N}$, where $N$ is the sample size, $x_i$ denotes the $i$-th instance and its label $y_i \in R^C$, where $C$ is the number of classes. Let the number of training data belonging to $k$-th class be $n_k$. Without loss of generality, we suppose that the classes are sorted in decreasing order, based on the number of training data in each class, i.e., $n_1 \geq ... \geq n_K$. Afterward, all classes can be recognized into two parts: head classes (referred as $G_h$) and tail classes (referred as $G_t$).

As shown in Figure 1 (a), in previous approaches, the model uses a vision encoder to map the input image $x_i$ to a d-dimensional vector $\mathbf{z}_i$. The vector $\mathbf{z}_i$ is usually converted to the class logits by a classifier. The output of the model can be defined as:

$$f(x_i) = h(\varepsilon_{vis}(x_i)) = h(\mathbf{z}_i), \tag{1}$$

The model parameters are trained by minimizing supervised loss function, such as cross-entropy, or LACE(Cao et al., 2019a) loss when the training data is imbalanced.

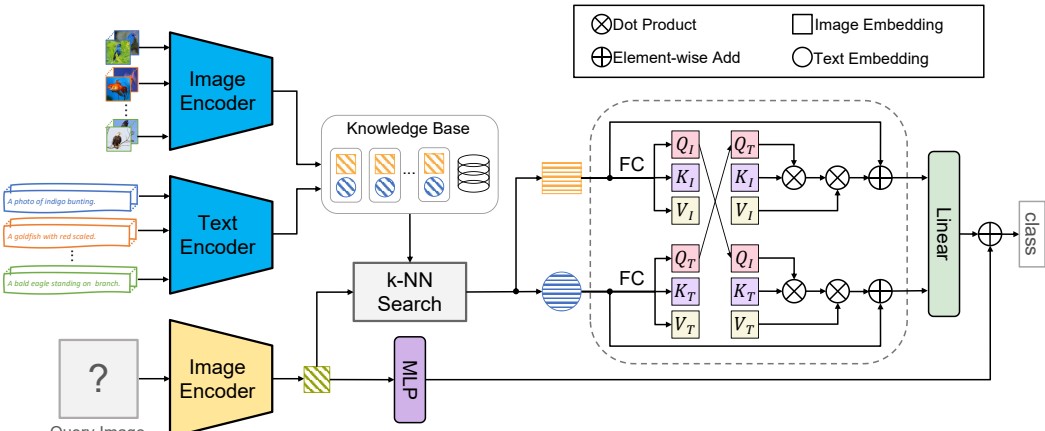

Figure 2: **Overall architecture of RetFormer** First, the image encoder will map the query image to an embedding. The image embedding will go through two branches, in the first branch the embedding goes through a simple MLP, in the second branch the image embedding first retrieves the relevant information from the knowledge base and then the contribution of each part is calculated by the retrieval cross-fusion module. The outputs of both branches are merged and trained by the same loss function.

## 3.2 RETFORMER

Retrieving from an external knowledge base enables RetFormer to incorporate external knowledge to enhance its performance. We describe the retrieval augmented classification and each of RetFormer's components in detail in this section.

### 3.2.1 RETRIEVAL AUGMENTED CLASSIFICATION

Typically, a classification model is trained in a downstream task to make predictions considering only the image $x_i$ in the dataset. The image $x_i$ is passed through the vision encoder to produce image representations and ultimately output class logits by the classifier.

Retrieval augmented classification aims to train more robust and accurate models by utilizing relevant information from the external knowledge base. We introduce an additional knowledge base $D = \{(I_i, T_i)\}_{i=1}^L$, consisting of images $x_i$ and corresponding labels $y_i$ in the dataset. More specifically, in addition to $x_i$, the model prediction now depends on $D$. Note that $D$ is independent of $S$, which means that $D$ is a database containing additional world knowledge, so we do not assume that $D$ contains the class labels of $S$.

For predicting the logits of a given image $x_i$, we construct a subset of $D$ that is most relevant to $x_i$ for improving the performance of the network. Then, two frozen encoders $\varepsilon_I$ and $\varepsilon_T$ act on the image and text in $D$ to convert them into embeddings as Eqn. 2, where $\varepsilon_I$ is vision encoder and $\varepsilon_T$ is text encoder:

$$E_i^I = \varepsilon_I(I_i), E_i^T = \varepsilon_T(T_i), \tag{2}$$

Let $V_D = \{(E_i^I, E_i^T)\}_{i=1}^L$ be the set of feature embeddings of each pair of instances in $D$. We compute the cosine similarity between $\mathbf{z}_i$ and each embedding $E^I \in V_D$ to find the k-nearest neighbors. The top-k ranked embeddings and query image embedding are then used for the output logits:

$$f(x_i) = \frac{\tau}{2}(L_I + L_R) = \frac{\tau}{2}(MLP(\mathbf{z}_i) + h(r(\mathbf{z}_i, V_{NN(\mathbf{z}_i;V_D)}))), \tag{3}$$

Here, $L_I$ and $L_R$ are the logits based on the query image and retrieval module, respectively. $MLP(\cdot)$ denotes two linear layers with a ReLU in the middle. $r(\cdot, \cdot)$ is a retrieval module and will be discussed in Section 3.2.2. $V_{NN(\mathbf{z}_i;V_D)}$ denotes top-k ranked embeddings of $\mathbf{z}_i$ from $V_D$.

### 3.2.2 RETRIEVAL CROSS-FUSION MODULE

The vectors in $V_{NN}$ are multi-classes and multi-modal, so it is important to model the relationship between each vector in the $V_{NN}$ and the query vector. We propose to compute the attentional weights between the query vector $\mathbf{z}_i$ and the retrieved $V_{NN}$ to represent the above relationship. These vectors are mapped from the CLIP encoders into the same feature space to learn the contribution of each vector in the $V_{NN}$.

First, we concatenate the vector $\mathbf{z}_i$ with the vectors in the $V_{NN}$ to get image embeddings $E_{NN}^I \in R^{P \times D}$ and text embeddings $E_{NN}^T \in R^{P \times D}$ (note that the text embedding corresponding to $\mathbf{z}_i$ is replaced with a $\mathbf{0}$ vector to prevent data leakage). The image embeddings $E_{NN}^I$ is mapped to three image matrices: query matrix $Q_I$, key matrix $K_I$ and value matrix $V_I$ by three linear transformations. The text embeddings $E_{NN}^T$ is mapped to three image matrices: query matrix $Q_T$, key matrix $K_T$ and value matrix $V_T$ by three linear transformations:

$$Q_I = E_{NN}^I W_{Q1}, K_I = E_{NN}^I W_{K1}, V_I = E_{NN}^I W_{V1}, \tag{4}$$
$$Q_T = E_{NN}^T W_{Q2}, K_T = E_{NN}^T W_{K2}, V_T = E_{NN}^T W_{V2},$$

where $W_{Q1}, W_{Q2}, W_{K1}, W_{K2}, W_{V1}$ and $W_{V2} \in R^{D \times D}$.

The outputs of retrieval cross-fusion module are represented as follows:

$$Att(Q, K, V) = \sigma(\frac{QK^T}{\sqrt{d}})V, \tag{5}$$
$$r(\mathbf{z}_i, V_{NN(\mathbf{z}_i; V_D)}) = Att(Q_T, K_I, V_I) + E_{NN}^I, Att(Q_I, K_T, V_T) + E_{NN}^T,$$

where $\sigma(\cdot)$ denotes Softmax function. $\frac{1}{\sqrt{d}}$ is the scaling factor for appropriate normalization to prevent extremely small gradients.

Note that Eq (5) can be repeated $L$ times, i.e. $L$ layers. Let $E_1^I, E_1^T = r(\mathbf{z}_i, V_{NN(\mathbf{z}_i; V_D)})$ denote the output of the first layer. The output after $L$ layers can be computed as:

$$E_L^I = Att(Q_{L-1}^T, K_{L-1}^I, V_{L-1}^I) + E_{L-1}^I, \tag{6}$$
$$E_L^T = Att(Q_{L-1}^I, K_{L-1}^T, V_{L-1}^T) + E_{L-1}^T,$$

In addition, we add Position embedding and class token references to ViT's(Dosovitskiy et al., 2020) setting prior to the input. Similar attention mechanisms are used for different purposes, such as feature fusion(Chen et al., 2021). Our experiments show that this attention mechanism is a good choice for retrieval augmented, significantly outperforming other baselines.

### 3.2.3 KNOWLEDGE BASE

The knowledge base is an important factor in the performance of the retrieval module, and we will now describe in more detail the datasets that make up the knowledge base. The impact of different choices on performance will be evaluated.

**Downstream dataset.** Building a knowledge base directly using the downstream dataset is the most straightforward option. This guarantees that for each query image, there will exist at least one instance of the same class. The disadvantage of this choice is that most downstream datasets are not rich enough in textual descriptions and may contain only the labels of the images.

**DataComp.** DataComp(Gadre et al., 2024) is a testbed for dataset experiments centered around a new candidate pool of 12.8 billion image-text pairs from Common Crawl. We use the subset of DataComp, which is the output of the Image-based and CLIP score baseline filter at the xlarge scale of DataComp and comprised of 1.4B samples.

**All.** To integrate world knowledge, we combined all of the above datasets, and there are about 1.4 billion image-text pairs in the knowledge base.

**Vectorize Knowledge base.** To extract features and compress storage, We initialize the encoder using the parameters of the CILP(Radford et al., 2021) pre-trained on DataComp-1B(Gadre et al., 2024) and freeze the parameters to the vectorize knowledge base prior to training and validation. This choice saves significant overhead and allows us to efficiently use in-memory datasets with up to 1B images. We will calculate the complexity of RetFormer in detail in Section 3.2.4.

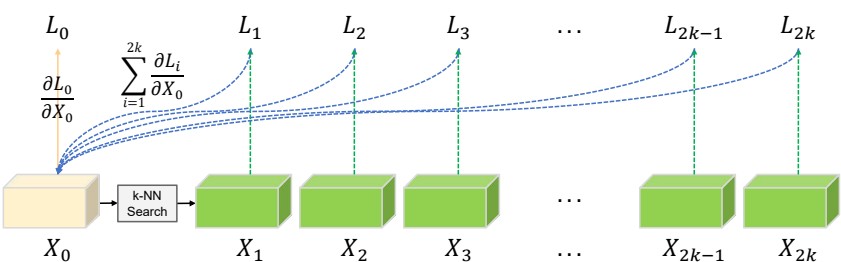

Figure 3: **Explaining RetFormer in terms of gradient propagation.** Blue dashed lines represent new gradient propagation among samples.

### 3.2.4 RETRIEVAL COMPLEXITY

**Time complexity.** We use the HNSW(Malkov & Yashunin, 2018) algorithm from the Faiss(Johnson et al., 2019) library to perform approximate k-NN searches in our experiments. It has sub-linear complexity, which means that for an in-memory dataset of $N$ elements, it takes $O(logN)$. In fact, the time consumed for querying in a 1B-element knowledge base is on the order of milliseconds. Since the encoder parameters are frozen, the results of one precomputation can be reused in subsequent training and validation, which also saves a lot of computation.

**Spatial complexity.** We use a fixed vision encoder in our experiments, so we pre-calculated and saved the results for all k-NN, thus saving storage space. For example, ImageNet-LT contains 115.8K images from 1000 categories. In our experiments, you need about 30GB of extra space for training.

### 3.2.5 A GRADIENT VIEW OF RETFORMER

To better understand how RetFormer helps representation learning by exploring sample relationships, we also provide an intuitive explanation from the perspective of gradient propagation. Intuitively, without the retrieval module, all losses would only propagate gradients over the corresponding samples and categories, i.e., one-to-one. Whereas, in our approach, there are gradients on other samples in the retrieval module, as shown in Figure 3. Specifically, given a sample $x_0$ with a retrieval subset $D_{NN} = \{(I_{2i-1}, T_{2i})\}_{i=1}^{k}$ of size $k$ and corresponding losses $L_0, L_1, L_i, ..., L_{2 \times k}$, we have:

$$\frac{\partial L_0}{\partial x_0} := \frac{\partial L_0}{\partial x_0} + \sum_{i=1}^{2k} \frac{\partial L_i}{\partial x_0}. \tag{7}$$

That is, the retrieval module brings in new gradient terms $\frac{\partial L_i}{\partial x_0}$. From a perspective of gradient optimization, $L_i$ also optimizes the network according to samples $(I_{2i-1}, T_{2i})\}_{i=1}^{k}$, which is significantly different compared to the model without the retrieval module. In other words, $(I_{2i-1}, T_{2i})\}_{i=1}^{k}$ can be thought of as virtual samples of $y_i$, where $y_i$ is the label of $x_i$. We believe that the retrieval module can be considered as a data-dependent augmentation. The retrieval module implicitly extracts virtual samples and models multimodal inter-sample relationships by means of samples in their neighborhood distributions. From this perspective, the retrieval module implicitly adds virtual samples for each label $y_i$ by modeling the relationship between samples in a k-NN search. Virtual samples are useful for tail classes because these classes lack samples. Previous approaches(Balaji et al., 2018; Zhu et al., 2018) have shown that data augmentation helps in long-tailed recognition and learning with noisy labels.

## 4 EXPERIMENTS

### 4.1 DATASETS AND IMPLEMENTATION DETAILS

We focus on two different image classification tasks: long-tailed recognition and learning with noisy labels. We now describe the downstream datasets we used for each task.

Table 1: Test top-1 accuracy (%) on CIFAR-100-LT with varying imbalance ratios. Retformer outperforms prior arts when using a similar backbone network. Pre-trained model from ImageNet-21K has several classes related to CIFAR-100(Krizhevsky et al., 2009), which potentially leads to data leakage.

| Method | Extra Data | Backbone | imbalance ratio | | |
|---|---|---|---|---|---|
| | | | 100 | 50 | 10 |
| Training from scratch | | | | | |
| BCL(Zhu et al., 2022) | × | ResNeXt-50 | 51.93 | 56.59 | 64.87 |
| GLMC(Du et al., 2023) | × | ResNeXt-50 | 55.88 | 61.08 | 70.74 |
| SURE(Li et al., 2024) | × | ResNet32 | 51.60 | 58.57 | 71.13 |
| Fine-tuning pre-trained model | | | | | |
| LiVT(Xu et al., 2023) | × | ViT-B/16 | 58.2 | 82.0 | 69.2 |
| BALLAD(Ma et al., 2021) | ✓ | ViT-B/16 | 77.8 | - | - |
| PEL(Shi et al., 2023) | ✓ | ViT-B/16 | 80.3 | 82.0 | 83.8 |
| Ours | ✓ | ViT-B/16 | **81.4** | **83.0** | **84.5** |
| Fine-tuning pre-trained model from ImageNet-21K | | | | | |
| LPT(Dong et al., 2022) | ✓ | ViT-B/16 | 89.1 | 90.0 | 91.0 |
| PEL(Shi et al., 2023) | ✓ | ViT-B/16 | 89.1 | 90.2 | 91.3 |

**Long-tailed Learning.** We use two datasets for Long-tailed learning: CIFAR-100-LT and ImageNet-LT. The CIFAR-100-LT is derived from the CIFAR-100(Krizhevsky et al., 2009) with constructed imbalance ratios including 10, 50, and 100. ImageNet-LT has 1000 classes, each with a number of training images ranging from 5 to 1280. It is created by acquiring a subset of the original ImageNet dataset, so the number of images per class follows a long-tailed distribution.

**Learning with noisy labels.** For Learning with noisy labels. We trained RetFormer on WebVision and tested on the WebVision and ILSVRC12 validation set. WebVision contains 2.4 million images crawled from the website using the 1000 concepts shared with ImageNet ILSVRC12. Following the "mini" setting in (Ma et al., 2020), we take the first 50 classes of the Google resized image subset. We then test the trained network on the same 50 classes of the WebVision and ILSVRC12 validation set.

**Implementation details.** To avoid data leakage, we initialized the vision encoder using the ViT-B/16 parameters of CILP pre-trained on DataComp-1B instead of ImageNet-21K. Unless otherwise stated, training lasted for 25 epochs with a learning rate of 0.0005 and batch size of 256. the learning rate followed a 1-epoch warm-up schedule and then decreased at each epoch using a cosine decay schedule. We use the Adam optimizer(Kingma & Ba, 2014) with a weight decay of 0.1. we also use label smoothing(Szegedy et al., 2016) and Mixup(Zhang et al., 2017) during training to prevent overfitting and improve model generalization. For the attention module, we use $L = 4$. We retrieve $k = 32$ examples from memory unless otherwise stated.

## 4.2 RESULTS ON CIFAR-100-LT

Table 1 shows the results for CIFAR-100-LT. The results clearly show that RetFormer outperforms other methods including PEL, LiVT, BALLAD, and various ab initio training methods. Our method is the best among all methods that use extra data, demonstrating the potential of retrieval-augmented in vision tasks. The advantage of our method is more obvious when the dataset is unbalanced, which is due to the fact that the retrieval module makes the model focus on the tail classes appropriately. In addition, we do not compare models using ViT pre-trained on the ImageNet-21K dataset due to data leakage that would introduce unfair comparisons.

## 4.3 RESULTS ON IMAGENET-LT

We compare our method with the state-of-the-art methods on ImageNet-LT. Table 2 illustrates the top-1 accuracy of existing methods on ImageNet-LT. Note that the pre-training of ViT-B/16 is dif-

Table 2: Test top-1 accuracy (%) on ImageNet-LT. The best rusults are in bold for comparison. Partial numerical results come from Iscen et al. (2023).

| Method | Extra Data | Backbone | Many-shot | Med-shot | Few-shot | All |
|---|---|---|---|---|---|---|
| | | Training from scratch | | | | |
| PaCo(Cui et al., 2021) | × | ResNext-101 | 68.2 | 58.7 | 41.0 | 60.0 |
| LiVT(Xu et al., 2023) | × | ViT-B/16 | 73.6 | 56.4 | 41.0 | 60.9 |
| | | Fine-tuning pre-trained model | | | | |
| BALLAD(Ma et al., 2021) | × | ViT-B/16 | 79.1 | 74.5 | 69.8 | 75.7 |
| VL-LTR(Tian et al., 2022) | ✓ | ViT-B/16 | 84.5 | 74.6 | 59.3 | 77.2 |
| PEL(Shi et al., 2023) | ✓ | ViT-B/16 | 81.3 | 77.4 | 73.4 | 78.3 |
| RAC(Long et al., 2022) | ✓ | ViT-B/16 | 80.9 | 76.0 | 67.5 | 76.7 |
| MAM(Iscen et al., 2023) | ✓ | ViT-B/16 | 80.6 | 77.5 | 74.5 | 78.3 |
| Ours | ✓ | ViT-B/16 | **85.0** | **80.9** | **76.8** | **81.9** |

Table 3: Top-1 and top-5 test accuracy on WebVision and ImageNet validation sets. Partial numerical results come from Zhang et al. (2023).

| Train | mini-WebVision | | | |
|---|---|---|---|---|
| Validate | WebVision | | ILSVRC12 | |
| Method | Top1 (%) | Top5 (%) | Top1 (%) | Top5 (%) |
| HAR(Cao et al., 2020) | 75.5 | 90.7 | 70.3 | 90.0 |
| RoLT+(Wei et al., 2021) | 77.64 | 92.44 | 74.64 | 92.48 |
| NGC(Wu et al., 2021) | 79.16 | 91.84 | 74.44 | 91.04 |
| RCAL+(Zhang et al., 2023) | 79.56 | 93.36 | 76.32 | **93.68** |
| Sel-CL+(Li et al., 2022) | 79.96 | 92.64 | 76.84 | 93.04 |
| Dynamic Loss(Jiang et al., 2023) | 80.12 | 93.64 | 74.76 | 93.08 |
| Ours | **81.7** | **94.1** | **77.3** | 93.2 |

ferent between the methods. RAC and MAM use the ViT-B/16 vision encoder pre-trained on the JFT-3B(Zhai et al., 2022) and WebLI(Chen et al., 2022). BALLAD, VL-LTR, and PEL use the same ViT-B/16 pre-trained with CLIP as our method.

We see that VL-LTR achieved the high accuracy in Many-shot. However, our method achieved the overall highest accuracy due to the proper focus on the tailed classes. Our method outperforms other methods that also utilize extra data. RAC and MAM use retrieval-augmented as well, but they only consider the relationships between image modalities, which leads to suboptimal performance.

## 4.4 RESULTS ON WEBVISION

Table 3 shows the results on WebVision. It can be seen that our method achieves the best results on the top 1 accuracies on both the WebVision validation set and the ImageNet ILSVRC 12 validation set compared to other state-of-the-art methods. RCAL uses representations learned from unsupervised comparative learning, restores the underlying representation distribution, and then samples data points to balance the classifier.RCAL+ combines semi-supervised learning algorithms, resulting in slightly higher top5 accuracy on ImageNet than our approach.

## 4.5 ABLATION STUDY

In this section, we provide an in-depth analysis of RetFormer, where we compare the cross-fusion module with different baselines in order to demonstrate the advantages of the cross-fusion module. Unless otherwise stated, all settings are the same as in Section 4.1.

**Baseline settings.** We report the accuracy of the following baselines. CLIP zero-shot indicates the zero-shot performance of CLIP on the dataset. CLIP full FT means fine-tuning all the parameters of

Table 4: Comparing top-1 accuracy (%) with baselines on ImageNet-LT

| Method | Many-shot | Med-shot | Few-shot | All |
|---|---|---|---|---|
| CLIP zero-shot | 69.2 | 67.6 | 67.7 | 68.3 |
| CLIP full FT | 84.3 | 73.1 | 52.9 | 74.6 |
| CLIP classifier FT | 77.3 | 73.3 | 64.2 | 73.6 |
| Ours w/o text | 81.3 | 74.8 | 65.9 | 76.0 |
| Ours w/o image | 83.1 | 76.9 | 68.4 | 78.1 |
| Ours w/ FE | 79.3 | 79.0 | 74.6 | 78.6 |
| Ours | **85.0** | **80.9** | **76.8** | **81.9** |

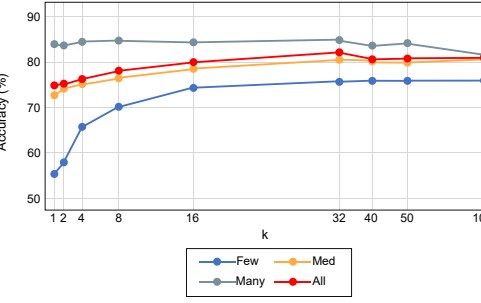 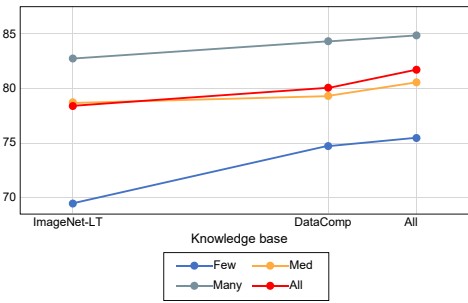

Figure 4: **Ablation study on ImageNet-LT. Left:** We show the effect of different k on RetFormer. **Right:** We show the impact of building a knowledge base with different datasets, we set $k = 32$ in this experiment.

CLIP directly on the training set. CLIP classifier FT means fine-tuning only the parameters of the classifier. Ours w/o text will retrieve only similar images and ignore the corresponding descriptive text. Ours w/o image will retrieve only similar images but input only the corresponding descriptive text. In these two settings, our cross-fusion module will degenerate into a vanilla transformer encoder. Ours w/ FE means that the parameters of the image encoder are frozen during training.

Table 4 demonstrates the results of the ablation study. The results show that our retrieval module can effectively improve the accuracy of the tail class. This can be proved by the performance of CLIP full FT. On the other hand, retrieving the image alone gives a small performance improvement, while retrieving the text corresponding to the image improves the overall accuracy. The performance of Ours w/ FE is explained in Section 3.2.5.

**Effect of k.** The impact of $k$ on model performance cannot be ignored. This hyperparameter controls the number of samples retrieved from the knowledge base. As illustrated in Figure 4 (left), we see that the performance gradually increases up to $k = 32$ and stabilizes thereafter. The performance of Many-shot decreases instead as $k$ increases, which may be due to the fact that the sample size of Many-shot is sufficient for the model to find the decision boundary, and the noise labels introduced by too large $k$ reduces the performance instead. The sample size of Few-shot is gradually stabilized as $k$ increases, which is consistent with our theoretical analysis in Section 3.2.5.

**Impact of Knowledge base.** We will continue to analyze the impact of selecting different datasets to build a knowledge base on model performance. Figure 4 (right) shows the performance of building a knowledge base using different datasets. In this experiment, we set $k = 32$. we see that the performance of the model gradually increases as the size of the knowledge base increases. Among them, the rise of the Few-shot accuracy is greater, which is in line with our theoretical analysis.

**Index ablations.** We examine the effect of the index type and recall on RetFormer's performance and speed in Table 5. To quantify error induced by an approximate index, we include the accuracy on the index content itself in addition to the validation accuracy. We observe that the drop in accuracy due to use of an approximate (HNSW) instead of exact k-NN is also minor, but comes with a significant (3×) speedup on large index's. In summary, the choice of HNSW becomes critical to ensure lookup time does not bottleneck training.

Table 5: **Index ablations on ImageNet-LT.** Recall represents the recall rate of the approximate k-NN algorithm under different settings. QT indicates Query Time.

| Index Type | Distance | Recall | Many-shot | Med-shot | Few-shot | All | QT (ms/sample) |
|---|---|---|---|---|---|---|---|
| Exact | Cosine | 1.00 | 85.0 | 80.9 | 76.8 | 81.9 | 23.34 |
| HNSW | Cosine | 0.97 | 85.1 | 80.7 | 76.5 | 81.8 | 7.69 |
| HNSW | Cosine | 0.91 | 84.8 | 79.9 | 74.9 | 81.1 | 5.29 |
| HNSW | Cosine | 0.82 | 84.2 | 79.0 | 73.1 | 80.2 | 3.47 |
| HNSW | Cosine | 0.65 | 83.1 | 77.7 | 69.7 | 78.7 | 2.41 |

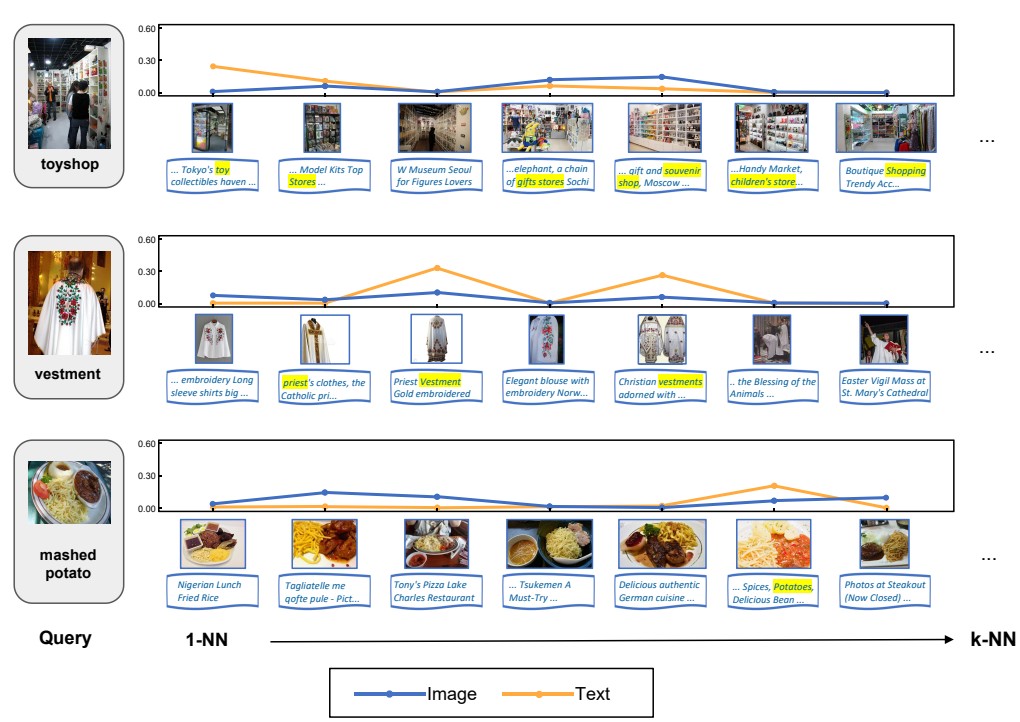

Figure 5: **Qualitative Example.** We visually demonstrate how the retrival cross-fusion module copes with tailed classes. Query images of tailed classes are shown on the left. The k-NN images from the knowledge base are shown on the right and sorted from left to right. We display the attention weight assigned to each k-NN above the corresponding image.

**Qualitative examples.** We present some of the qualitative examples in Figure 5. We observed that our method assigns higher attention weights to both relevant images and text in the k-NN list, indicating that RetFormer can capture effective relationships from two different modalities. We found that even in the absence of any relevant images, our method still benefits from related shared knowledge.

## 5 CONCLUSION

In this work, we introduce RetFormer, a new multimodal retrieval-augmented vision language framework for long-tailed recognition and learning with noisy labels. We emphasize that the image and text modality of retrieved instances have implicit intrinsic relationships, and propose to enable deep neural networks themselves with the ability to explore these sample relationships. We propose a simple, but effective, retrieval cross-fusion module that learns multimodal sample relationships and computes their contributions. Extensive experiments on a variety of long-tailed recognition and learning with noisy labels benchmarks validate that our approach works better than well-designed vision-based methods.

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
