# OpenReview forum: "RetFormer: Enhancing Multimodal Retrieval for Image Recognition"
_ICLR.cc/2025/Conference — ICLR 2025 Conference Withdrawn Submission_

### Official Review · Reviewer_Gc1N · 2024-10-17

**Soundness:** 1
**Presentation:** 2
**Contribution:** 2
**Rating:** 3
**Confidence:** 4

**Summary:**

The authors use an external image-text pair knowledge base to store world knowledge and then use a retrieval module to identify and obtain relevant knowledge from a predefined knowledge base.
The work intends to tackle real-world problems like
1. “Data Scarcity Learning” (Learning with noisy labels)
2. “Long-tailed recognition”
3. “Efficiency” Learning externally without gain in model parameters
The method overcomes the limitations of traditional methods on model size and training datasets by seamlessly integrating information from external databases into the model’s decision-making process.

**Strengths:**

1. The proposed method of retrieval-based solution is justified  [Line 92-95] “overcome the limitations of traditional methods on model size and training datasets by seamlessly integrating information from external databases into the model’s decision-making process.”

2. Work shows impressive performance on three benchmark datasets.

3. The ablation study describes the hyper parameters nicely, K means clusters.

**Weaknesses:**

1. **Too many motivations are confusing**. Applying Large Scale Foundation models in real world is complicated because
         a.	long-tailed distribution problem (skewed distribution of samples/class (not mentioned in the paper)).
         b.	Noisy labels in the real world
         c.	There are too many parameters in the models for learning sample relationships. (efficiency issues in existing methodology)
         d.	Retrieval-based solutions are under-explored.
The solution explicitly answers only (c) and (d). At this point, authors are requested to justify how their methodology solves the "long-tailed distribution problem" and "Noisy labels in the real world" problem or may be re-worded to remove (a) and (b) as a motivation.

2. **Writing needs improvement**
       a.	[Line 45-48]  “challenges”, show or cite a paper.
       b.	[Line 73-74] “retrieval-augmented into the field of vision recognition” What does this mean?
       c.	[Line 081-083] “When we search for different classes of samples, there may be shared knowledge transfer in the image and text modalities with the query image.” (what does this mean?)
Please clarify what the above means (re-word) in the context of vision recognition, perhaps with an example or more precise definition.


3. **Unexplained Symbols**
      a.	[Line 198] D= I,T. What I and T? Show I,T in [Figure 2].
     c.	What is “h” in [equation 3]?
     d.	[Line 220 -221], where did CLIP come from? It's not mentioned anywhere before.
     e.	[Line 224], what P and D? `D' symbol is already used for the external knowledge source
Please provide a clear definition or explanation for each of these terms and symbols when they are first introduced in the paper.


4. **Unanswered Motivation**
a.	What specific part of the methodology makes the model robust to long-tail distribution and noisy labels? Why this method can’t simply be applied to normal Classification?
b.	Are authors claiming “retrieval-based methods” are more robust to long tail distribution than normal pretraining-based models?
c.	If the claim is that retrieval-based methods are more efficient, use a smaller backbone to show comparable performance to VIT-B/16. If backbones are as computation heavy and identical to existing works, how is efficiency being answered? What section of the methodology focuses on efficiency?
d.	Results seem to indicate “Pretraining on ImageNet” is better than DataComp-1B based pertaining. So pertaining is better than the retrieval-based solution.
e.	How is the proposed method more efficient than any previous method? Show GFLOP values and compare inference time/query.
Ablation needs to show all of these points, particularly focusing on how their method compares to traditional approaches in terms of robustness to long-tail distributions and efficiency.




5.	**Author criticizes the use of a model with additional parameters and large-scale training data [line 92-94]**, but uses additional dataset “D” and two image encoders.  Storing the CLIP model in memory or 1.4 billion embeddings in memory, none of it is efficient. Hosting two image encoders (that too transformers) restricts their application to very light methods.
a.	Data Comb is one of the largest pertaining datasets available (used in pretraining most of the foundation models). In addition to pretraining, the authors are using a 1.4 B image-text pair as an additional dataset for retrieval. How is this method more efficient than traditional pertaining methods?
Please clarify how using large datasets and multiple encoders, addresses the efficiency concerns they raised about other methods. Provide a detailed comparison of computational and memory requirements between their method and traditional pretraining approaches.

6.	**Outdated Works (2023 and earlier)** Most cited works seem to be majorly from 2023 and previous years. A simple paper with code search reveals missing cited works e.g.
a) (ImageNet-LT) Long-Tail Learning with Foundation Model: Heavy Fine-Tuning Hurts (ICML 2024)
(b) (ImageNet-LT) MAM (Iscen et al., 2023) results for fine-tuning is  85.4 and not 80.6. (Many-shot)
(c) (mini-WebVision) Scenic: A jax library for computer vision research and beyond. (CVPR'22)
(d) (mini-WebVision) Sample prior guided robust model learning to suppress noisy labels."
(e) (mini-WebVision) Label-Retrieval-Augmented Diffusion Models for Learning from Noisy Labels (Nuerips'24)


Additional Feedback
1.	Explain the terms, e.g. long-tailed recognition.
2.	Leave a space between text and citation bracket “(“, e.g. GPT-4(Achiam et al., 2023),  GPT-4 (Achiam et al., 2023),
3.	Figure ordering should reflect text description. It's weird to describe Figure 1 (c)(d)(e), [Line 079] before Figure 1 (b), [Line 88]

**Questions:**

Please answer all the weakness

---

### Official Review · Reviewer_Cb5T · 2024-10-23

**Soundness:** 2
**Presentation:** 2
**Contribution:** 2
**Rating:** 3
**Confidence:** 3

**Summary:**

This paper proposes a knowledge base-enhanced structure called RetFormer, which attempts to overcome poor performance caused by long-tailed and noisy data during training by integrating external knowledge retrieval and a cross-attention mechanism. The experimental results in this paper are promising, but retrieval-augmented everything is not a new story. The introduction of world knowledge (a subset of DataComp, 1.4B) essentially equates to indirectly increasing the data, so the improvement in accuracy is understandable and did not provide me with new insights.

**Strengths:**

1. The effectiveness of introducing retrieval augmentation is evident.

2. The experiments and analyses involving different scenarios, such as CLIP and retrieval, are relatively comprehensive

**Weaknesses:**

1. Figure 1 shows a case of noisy labels (spelled as "Lable" by the authors), but in the proposed method, this information is directly fused into the image features through cross-attention. How can classification be ensured to not be influenced by the noise in semantics?

2. The retrieval number k is set to 32 in this paper, using it for cross-attention may lead to significant GPU memory overhead. Moreover, in Figure 4, the improvement for Any-Shot from k = 16 to 32 is not very significant. Why not introduce clustering or prototype-based methods to further improve it?

3. There is a lack of citations for some related works on retrieval-augmented multimodal learning.

**Questions:**

Most of my concerns are outlined in the Disadvantages section. Below are some works that may need to be referenced (though they are not directly related to image classification tasks, I am not sure if these should be additionally cited).
[1] REVEAL: Retrieval-Augmented Visual-Language Pre-Training with Multi-Source Multimodal Knowledge Memory (CVPR2022)
[2] Fine-grained Late-interaction Multi-modal Retrieval for Retrieval Augmented Visual Question Answering (NeurIPS2023)
[3] Retrieval-Augmented Multimodal Language Modeling (ICML2023)

In addition, I have some questions:

1. In lines 288-289, it is mentioned that "the encoder parameters are frozen, and the results of one precomputation can be reused in subsequent training and validation." Does this mean that retrieval is not conducted online?

2. For each query image, are the samples introduced by kNN fixed? If so, why not directly add this part of the data for pre-training? I suspect that this would produce a similar effect.

3. Line 473 mentions decision boundaries. I am curious if it is possible to visualize the feature changes before and after introducing additional CLIP features through kNN, and how it affects the decision boundaries. Why does retrieval augmentation improve performance on long-tailed and noisy data? Is it all due to the extra data, or does cross-attention play a role as well?

---

### Official Review · Reviewer_4LXg · 2024-10-26

**Soundness:** 3
**Presentation:** 3
**Contribution:** 2
**Rating:** 5
**Confidence:** 3

**Summary:**

The authors proposed a novel multimodal retrieval architecture Retformer for long-tail learning and noisy label learning. Specifically, it establishes a robust relationship between image and text modalities by integrating information from an external knowledge base into the model's decision process, thereby overcoming the limitations of traditional methods on model size and datasets. In addition, the authors proposed a cross-fusion module to leverage the content in the knowledge base to establish robust multimodal sample relationships. Experiments have demonstrated its effectiveness.

**Strengths:**

1) The motivation that leverages content from the knowledge base to establish multimodal sample relationships for enhancing visual tasks is interesting and reasonable.
2) The method has achieved new state-of-the-art results.
3) The paper is well-written and easy to follow.

**Weaknesses:**

1) The real world often follows an imbalanced distribution and contains noisy labels. The author only verifies the effectiveness of Retformer on the two tasks separately. I am interested the whether Retfomer can handle long-tailed noisy learning [a].
2) Lack of novelty. Multimodal Retrieval has already been explored with previous methods such as MAM.  The basic idea and method are very similar.
3) Why the Reformer can handle long-tailed learning or noisy label learning? Normally, long-tailed learning aims to solve the underrepresentation of tail classes while noisy label aims to solve the samples with the wrong label. How does the proposed method solve these abnormal data?
4) For long-tailed learning, you should also report the performance on Places-LT and iNaturalist2018.

[a] When Noisy Labels Meet Long Tail Dilemmas: A Representation Calibration Method. ICCV2023

**Questions:**

see weakness

---

### Official Review · Reviewer_VYXa · 2024-10-28

**Soundness:** 2
**Presentation:** 2
**Contribution:** 1
**Rating:** 3
**Confidence:** 5

**Summary:**

This work introduces RetFormer, one retrieval-augmented method for long-tailed image recognition. It uses one multi-modal knowledge base and one retrieval cross-fusion to establish sample relationships between image and text modalities. Some experiments training with noisy labels benchmarks on long-tailed recognition are carried out for the claimed validation.

**Strengths:**

[+] Pursuing better interpretability is interesting for the current black box deep-learning paradigm.

[+] The paper is easy to follow and understand, having clear formulation.

[+] Some experiments are conducted to demonstrate the idea’s performance, as well as the values of modeling sample relationship.

**Weaknesses:**

[-] The "chicken or the egg" problem. To obtain good results for Retformer, one reliable image-text pair knowledge base (usually large-scale) is critical, which means a significant amount of external labor is required. On the one hand, it’s definitely unfair to directly compare the performance under different data. For example, the compared methods should also be training with the data from knowledge base. On the other hand, if we already have a good retrieval model/base that can be used to mine hard samples in Retformer, it seems unnecessary to train a classification model (because retrieval is actually an open set classification)?

[-] Insufficient comparisons. To model sample relationships, this paper uses the image-text retrieval-augmented pipeline, such that the model can see more hard positive/negative samples. While to achieve the goal, one vanilla solution is full data augmentation, e.g., rotation, mix up of images, and re-caption of texts (by LLMs). More experiments and discussions are needed.

[-] External knowledge base. It seems that the construction of knowledge base requires many assumptions or tricks, e.g., building a knowledge base directly using the downstream dataset (in Line255-258). And as shown in Fig. 4, the knowledge base has a significant impact on performance, which raises questions about the generalization/usability of Retformer ideas in practical scenarios.

**Questions:**

[-] Complexity. Comparing to directly training pure Transformers, Retformer introduces complex retrieval operations, thus it’s necessary to quantify the time and spatial complexity in Tab. 1-4.

---

### Official Review · Reviewer_a3KF · 2024-11-02

**Soundness:** 2
**Presentation:** 2
**Contribution:** 1
**Rating:** 3
**Confidence:** 4

**Summary:**

The paper introduces "RetFormer" a multimodal retrieval-augmented model designed to improve image recognition by leveraging both image and text information from an external knowledge base. This model addresses challenges in long-tailed distribution and noisy labels, which are common in real-world data. RetFormer claims enhance robustness by modeling relationships between image and text samples, demonstrating improved accuracy across benchmarks like CIFAR-100-LT, ImageNet-LT, and WebVision. In summary, the motivation behind the paper lacks clarity, the contribution should be better emphasized with respect to the existing works, experimental comparison should be fair and presentation of the paper should be improved.

**Strengths:**

The paper claims to achieve state-of-the-art results across various settings; however, in my opinion, the comparison is unfair, as noted in my comments in the weakness section.

**Weaknesses:**

(1) The motivation of the paper is not entirely clear to me. I understand that having a large number of model parameters can complicate the learning process and potentially lead to challenges, but I am unsure how it specifically limits interpretability and scalability. Additionally, how does a retrieval-augmented classification model address these issues?

(2) I believe that multimodal retrieval augmentation is already well-explored in the context of long-tail visual recognition. I suggest the authors review, discuss, and compare with the following published papers, among others. Currently, it is challenging to identify clear contrasts with existing works based on the presentation of the paper.
(a) Xie et al., RA-CLIP: Retrieval Augmented Contrastive Language-Image Pre-training, CVPR, 2023.
(b) Liu et al., Learning Customized Visual Models with Retrieval-Augmented Knowledge, CVPR, 2023.

(3) The paper's presentation could be improved. For instance, the first two points in the contributions section seem to refer to the same technical aspect. Additionally, Figure 1 includes two image encoders, but it is unclear how they are related and how are they used.

(4) The experiments section compares RetFormer with methods that use basic transformers with single-modal retrieval augmentation and have not incorporated multimodal retrieval augmentation in their training strategy. I believe this makes the comparison unfair.

**Questions:**

See the weaknesses section

---

### Official Review · Reviewer_9CS8 · 2024-11-03

**Soundness:** 1
**Presentation:** 1
**Contribution:** 1
**Rating:** 1
**Confidence:** 4

**Summary:**

This paper unveils a new model RetFormer to enhance image recognition tasks. RetFormer leverages a multimodal knowledge base and a retrieval cross-fusion module, allowing it to establish robust relationships between images and text. It integrates external knowledge into its decision-making process, overcoming traditional limitations on model size and datasets.  The experiments demonstrate its prowess in long-tailed recognition and learning with noisy labels, achieving state-of-the-art accuracies.

**Strengths:**

The usage of retrieval augmented strategy for classification.

**Weaknesses:**

- The writing is very awful. Much important information is missing.
- What problem does the proposed method focus on? Long-tail classification, learning with noisy label or efficient classification with less parameters?
- The motivation of this paper is very unclear. Why the retrieval-augmented classification can tackle with the long-tail and noisy label problems?
- The existing methods have introduced retrieval argumentation into classification. The difference between the proposed method and the other methods is not discussed.
- How about we fine-tune the CLIP on the target small dataset? Maybe more effective and efficient. More discussion should be added.
- Whether the proposed method can only work well on long-tail or noisy datasets? Why not evaluate it on other large-scale datasets? In fact, ImageNet also has about 7% noisy labels.
- More long-tailed classification and noisy label learning methods should be compared.
- The experimental setting is not fair. The pre-trained image encoder of CLIP  is used to initial image encoder of the proposed method.

**Questions:**

Please see Weakeness.

---

### Note · Authors · 2024-12-12

**Comment:**

Withdrawal

**Withdrawal Confirmation:**

I have read and agree with the venue's withdrawal policy on behalf of myself and my co-authors.